# Long Pentraxin-3 Follows and Modulates Bladder Cancer Progression

**DOI:** 10.3390/cancers11091277

**Published:** 2019-08-30

**Authors:** Sara Matarazzo, Laura Melocchi, Sara Rezzola, Elisabetta Grillo, Federica Maccarinelli, Arianna Giacomini, Marta Turati, Sara Taranto, Luca Zammataro, Marianna Cerasuolo, Mattia Bugatti, William Vermi, Marco Presta, Roberto Ronca

**Affiliations:** 1Department of Molecular and Translational Medicine, University of Brescia, 25123 Brescia, Italy; 2Department of Pathology, Fondazione Poliambulanza Hospital, 25124 Brescia, Italy; 3Department of Obstetrics, Gynecology, and Reproductive Sciences, Yale University School of Medicine, New Haven, CT 06520, USA; 4School of Mathematics and Physics, University of Portsmouth, Hampshire PO1 3HF, UK

**Keywords:** long pentraxin-3, bladder cancer, FGF, FGFR, stemness, metabolism

## Abstract

Bladder tumors are a diffuse type of cancer. Long pentraxin-3 (PTX3) is a component of the innate immunity with pleiotropic functions in the regulation of immune response, tissue remodeling, and cancer progression. PTX3 may act as an oncosuppressor in different contexts, functioning as an antagonist of the fibroblast growth factor/fibroblast growth factor receptor (FGF/FGFR) system, rewiring the immune microenvironment, or acting through mechanisms not yet fully clarified. In this study we used biopsies and data mining to assess that PTX3 is differentially expressed during the different stages of bladder cancer (BC) progression. BC cell lines, representative of different tumor grades, and transgenic/carcinogen-induced models were used to demonstrate in vitro and in vivo that PTX3 production by tumor cells decreases along the progression from low-grade to high-grade advanced muscle invasive forms (MIBC). In vitro and in vivo data revealed for the first time that PTX3 modulation and the consequent impairment of FGF/FGR systems in BC cells have a significant impact on different biological features of BC growth, including cell proliferation, motility, metabolism, stemness, and drug resistance. PTX3 exerts an oncosuppressive effect on BC progression and may represent a potential functional biomarker in BC evolution. Moreover, FGF/FGFR blockade has an impact on drug resistance and stemness features in BC.

## 1. Introduction

Bladder cancer (BC) is one of the most common cancers in the world, its worldwide incidence being estimated in more than 2 million patients per year, with a death rate of 400,000/year (source NCI, SEER Cancer Statistics Review). At presentation, approximately 70% of urothelial carcinomas (UC) are low-grade superficial papillary/non-muscle invasive tumors (NMIBC) with a relatively benign prognosis; the remaining 30% of UC are diagnosed as advanced muscle invasive forms (MIBC) with generally poor outcomes [1]. The conventional treatment of NMIBC involves surgical resection and intravesical chemo- or immunotherapy. One of the major challenges in the management of these tumors is their propensity to recur, thus requiring frequent and lifelong surveillance and making superficial BC the most expensive and time-consuming malignancy to treat [2]. In addition, approximately 15% of superficial tumors progress to become invasive. Despite treatment with radical cystectomy, radiotherapy, and adjuvant or neoadjuvant chemotherapy, newly diagnosed invasive bladder tumors and superficial tumors that have progressed to invasion often metastasize and the five-year survival rate is poor (<50%). Considering all these aspects and the fact that the UC incidence is increasing, BC represents a significant public health issue.

The fibroblast growth factor/fibroblast growth factor receptor (FGF/FGFR) system can be altered in a portion of UCs, and various FGFs (including FGF1 and FGF2) are found in urines and tumor tissues of bladder cancer patients and are frequently expressed (FGF1/2/5 and 8) in human bladder carcinoma cell lines [3,4]. Moreover, FGFR1 is expressed at high levels in both invasive and noninvasive tumors [5] and its expression is associated with poor survival in patients with resected muscle invasive UC [6]. Dysregulated FGFR3, either through mutation, overexpression, or both, has been found in approximately 80% of noninvasive UC, and 54% of invasive UC express FGFR3 [7].

The soluble pattern recognition receptor long pentraxin-3 (PTX3) is a component of the innate immunity. PTX3 is a member of the pentraxin family, which is produced locally in response to inflammatory signals and exerts nonredundant functions in various physiopathological conditions, including cancer. PTX3 is involved in different aspects of tumor progression, including tumor cell proliferation, angiogenesis, metastatic dissemination, and cancer immune modulation [8,9,10]. PTX3 can be produced by both tumor and stromal cells, making the study of its impact on the tumor microenvironment particularly complex [11]. PTX3 expression is silenced through promoter hypermethylation in different types of cancers [10,12] and may act as an oncosuppressor by modulating tumor-associated inflammation [12] and/or by blocking protumor growth factors like various members of the FGF family [11]. Accordingly, PTX3 overexpression can inhibit FGF-driven epithelial-to-mesenchymal transition (EMT) and tumor/metastatic burden in melanoma models [13] and hampers tumor growth in preclinical models of prostate cancer and fibrosarcoma [8,10]. 

PTX3 represents a protective innate immunity component able to prevent urinary tract infections [14] and it has been proposed as a potential biomarker in the urine of BC patients [15]. Nevertheless, no data are available about the role of PTX3 in BC and its impact on the biology of neoplastic bladder cells.

In this study, we report that PTX3 is differentially expressed at different stages of BC. We demonstrate that PTX3 production by tumor cells decreases along the progression from low-grade to high-grade/MIBC, with an oncosuppressive impact on the biological features of BC growth, including tumor cell proliferation, motility, metabolism, stemness, and drug resistance.

## 2. Results

### 2.1. PTX3 Expression Is Related to Tumor Grade in BC Patients and Cell Lines

Preliminary data mining on Gene Expression Omnibus (GEO) database revealed that the expression levels of PTX3 in BC are lower than those found in normal bladders, and decrease along tumor progression from non-muscle invasive lesions (NMIBC)/carcinoma in situ (CIS) to invasive cancer (MIBC) (Figure 1A). 

Based on these observations, a cohort of 62 human BC samples, including five low-grade papillary NMIBCs, 17 high-grade NMIBCs, and 40 MIBCs, were examined for the presence of PTX3 by immunohistochemistry and scored for PTX3 positivity (i.e., no presence, mild positivity, or high positivity; see Appendix A). As shown in Figure 1B,C, 100% of low-grade BC samples were strongly positive for PTX3 immunoreactivity, whereas 47% of high-grade NMIBCs were characterized by a lower/mild immunoreactivity, and 62.5% of high-grade muscle invasive BC samples showed a low (37.5%) or absent (25%) PTX3 positivity. Thus, BC progression is paralleled by a progressive loss of PTX3 expression, raising the possibility that PTX3 might exert an oncosuppressive role in BC.

To validate this hypothesis, three prototypic tumor cell lines representing different grades of BC progression (including low grade/papilloma-like RT4 cells, grade II 5637 cells, and grade III/muscle invasive HT1376 cells) were characterized for PTX3 expression. Notably, all the cell lines express various components of the FGF/FGFR system at different levels (see Appendix A).

As shown in Figure 1D, Western blot and ELISA analyses revealed that PTX3 is highly expressed by RT4 cells, whereas intermediate and extremely low PTX3 levels were observed in 5637 and HT1376 cells, respectively. Accordingly, invasive bladder cancer HT1376 cell xenografts grew faster and expressed negligible levels of PTX3 following in vivo implantation in immune-compromised mice when compared to lower grade 5637 lesions and papilloma-derived RT4 grafts (Figure 1E). Interestingly, a characterization of the central energy metabolism of BC cells using the Seahorse Mito Stress Test demonstrated that HT1376 and 5637 cells show only slightly decreased oxygen consumption rates (OCR) at basal conditions when compared to RT4 cells. At variance, a strong reduction of maximal respiration, as expressed by their OCR, accompanied by a significant increase in the glycolytic capacity, as assessed by maximal extracellular acidification rate (ECAR), occurred in HT1376 and 5637 cells when compared to RT4 cells following addition of the mitochondria uncoupler FCCP or of the ATP synthase inhibitor oligomycin, respectively. These data point to a stage-dependent rewiring of the energy metabolism during BC progression (Figure 1F and Appendix A).

High methylation levels of the *PTX3* gene promoter have been reported in different mesenchymal cancer types [10,12]. On this basis, we assessed the methylation status of the enhancers/regulatory regions for the *PTX3* gene in BC cell lines [12]. Analysis by Methylated CpG Island Recovery Assay (MIRA) showed methylation only in the putative *PTX3* enhancer encompassing the exon-2, and levels of methylated CpG islands were higher in HT1376 and 5637 cells when compared with the very low levels of methylation in RT4 cells (Figure 1G). Accordingly, treatment with the methylation inhibitor 5-Aza-2-deoxycytidine (5-AZA) significantly increased the expression of PTX3 at mRNA and protein levels in both hyper-methylated HT1376 and 5637 cells, with no significant effects on RT4 cells (Figure 1H). 

Together, these data show an inverse correlation between BC grade/aggressiveness and PTX3 expression. On this basis, in order to understand the biological significance of the different levels of PTX3 expression in tumor cell lines representing different BC grades, PTX3 expression was silenced in RT4 cells and upregulated in 5637 and HT1376 cells, as well as in the highly aggressive murine BC MB49 cells.

### 2.2. PTX3 Silencing Enhances Aggressiveness of Low-Grade BC Cells

As shown in Figure 2A, short hairpin RNA (shRNA) transduction efficiently downmodulated PTX3 expression in RT4-shPTX3 cells when compared to control (RT4-shNT) and wild-type cells. 

In vitro characterization revealed that PTX3 silencing increased significantly the proliferative and clonogenic capacity of RT4-shPTX3 cells in respect to control RT4-shNT cells (Figure 2B,C). Moreover, RT4-shPTX3 cells showed an increased anchorage-independent growth capacity when seeded in soft agar (Figure 2D). These observations go along with an increased activation of the FGF/FGFR system in RT4-shPTX3 cells, as represented by the augmented phosphorylation of FGFR, FRS2 adapter, and mitogenic ERK1/2 proteins (Figure 2E).

Despite these differences in terms of proliferative potential, no significant difference was found for basal, ATP-linked, maximal OCR, ECAR, and ATP production (Figure 2F), indicating that modulation of PTX3 levels is not sufficient to induce significant changes in the energy metabolism of RT4 cells. 

In vivo, when grafted into immunocompromised mice, PTX3 downregulation resulted in increased tumor burden of RT4-shPTX3 lesions when compared to control and wild-type tumors, accompanied by an increased proliferation index as assessed by immunostaining for phospho-histone H3-positive (pHH3^+^) cells (Figure 2G). Moreover, immunostaining for classical basal (CK14, CK5/6, CD44) and luminal (CK20, UPK2) biomarkers [16] suggests that both control and PTX3-modulated RT4 cells maintain their histological features/markers in vivo (Appendix A).

### 2.3. PTX3 Overexpression Exerts an Oncosuppressive Effect on High-Grade BC Cells

As shown in Figure 3, 5637 cells expressed low levels of PTX3 and PTX3-overexpression (Figure 3A), which significantly impaired their proliferative (Figure 3B) and clonogenic (Figure 3C) potential when compared to control cells. Moreover, when tested for their capacity to repair a wounded monolayer, 5637-hPTX3 cells showed a reduced motility/wound repair capacity (Figure 3D). In addition, Seahorse Mito Stress Test revealed a significant reduction of cellular respiration and glycolysis in PTX3-overexpressing cells that were accompanied by a reduced production of ATP (Figure 3E).

As already shown for RT4 cells, the modulation of PTX3 expression impacted on the activation of the FGF/FGFR signaling pathway in 5637 cells. Indeed, Western blot analysis showed a reduced phosphorylation of FGFR and FRS2 in 5637-hPTX3 cells, which was followed by reduced ERK1/2 activation (Figure 3F). In addition, as a consequence of PTX3 upregulation, in vivo grafting in immune-deficient mice resulted in reduced tumor burden/growth capacity of 5637-hPTX3 cells in comparison with 5637-Mock and wild-type cells (Figure 3G). Immunohistochemical pHH3 analysis confirmed that 5637-hPTX3 tumors were characterized by a reduced proliferation rate (Figure 3G) with no changes in the expression of basal markers (Appendix A).

In muscle invasive HT1376 cells, forced expression of PTX3 (Figure 4A) hampered the clonogenic potential of HT1376 cells (Figure 4B) with no significant effect on their proliferation and wound repair features (Appendix A). Interestingly, as already observed for 5637 cells, PTX3-overexpression had a significant impact on the energy metabolism of HT1376-hPTX3 cells. As shown in Figure 4C (and Appendix A), HT1376-hPTX3 cells were characterized by a lower OCR with no changes in ECAR; this may explain the reduced ATP production in comparison with control HT1376-Mock cells. Again, Western blot analysis confirmed that PTX3 overexpression caused a reduction in FGFR and FRS2 activation (Figure 4D). In agreement with in vitro results, in vivo grafting of HT1376 transfectants in immune-deficient mice revealed that HT1376-hPTX3 cells have a mildly reduced tumor growth capacity when compared with control and wild type cells (Figure 4E) with similar expression of the cell proliferation marker pHH3 (Figure 4E).

To get further insights about the impact of PTX3 overexpression in MIBC, we took advantage of MB49 cells, a murine model of high-grade BC [17] that allows the growth of tumor cells in a syngeneic and immunocompetent background. In keeping with our observations on human HT1376 cells, MB49 cells have undetectable levels of PTX3 and the overexpression of PTX3 in MB49-hPTX3 transfectants (Figure 4F) caused a significant decrease in their proliferation (Figure 4G) and clonogenic/anchorage-independent growth capacity in soft agar (Figure 4H). This was accompanied by the reduction of FGFR, FRS2, and ERK1/2 activation/phosphorylation (Figure 4I). Accordingly, in vivo grafting of PTX3-overexpressing MB49 cells in syngeneic animals resulted in smaller tumors with a reduced Ki67^+^ cell proliferation index when compared to mock control and wild-type grafts (Figure 4J).

### 2.4. PTX3 Modulation Affects Stemness Features in BC Cells

As in other cancer types, the high frequency of recurrence and/or therapy failure in BC have been attributed to the presence of a subpopulation of undifferentiated/stem-like tumor cells [18]. To investigate whether the modulation of PTX3 expression might impact the stem-like behavior/population in low and high-grade BC, a sphere-forming assay was performed on PTX3 modulated RT4-shPTX3, 5637-hPTX3, and HT1376-hPTX3 cells and on their corresponding controls (RT4-shNT, 5637-Mock, and HT1376-Mock cells, respectively). As shown in Figure 5A, PTX3 silencing caused a significant increase of the sphere formation capacity of RT4 cells. Conversely, PTX3 overexpression hampered the capacity of 5637 and HT1376 cells to form spheres. Accordingly, flow cytometry evaluation of ALDH activity (a functional marker for stem-like cell populations) performed on disaggregated spheres revealed a significant increase in the percentage of ALDH^+^ cells in RT4-shPTX3 vs. RT4-shNT cells (6.34% vs. 2.92%). Conversely, a decrease of the ALDH^+^ cell population was observed in 5637-hPTX3 vs. 5637-Mock cells (22.38% vs. 15.19%) and in HT1376-hPTX3 vs. HT1376-Mock cells (11.51% vs. 5.18%) (Figure 5B). 

In keeping with these observations, an increase in the expression of the stemness markers *NANOG*, *OCT4*, *CD47* and of the stemness-related multidrug resistance genes *ABCG2* and *ABCB1* occurred in RT4-shPTX3 cells when compared to RT4-shNT cells (Figure 5C). Conversely, a significant reduction of the expression of these stemness-related genes was observed in HT1376-hPTX3 and 5637-hPTX3 cells when compared to their mock/control counterparts (Figure 5C). Interestingly, HT1376-hPTX3 cells were characterized also by a significant modulation of the basal urothelial stem cell marker *CD44* and by a strong modulation of the expression of *ABCG2* and *ABCB1* genes that goes along with an increased sensitivity to treatment with the classical chemotherapeutic methotrexate (Appendix A).

### 2.5. Stromal Expression of PTX3 Hampers BC Growth

Down- and upregulation expression experiments performed on BC cell lines indicated that PTX3 may exert an efficacious “tumor brake” effect in BC (see above). These observations prompted us to investigate whether a stromal overexpression of PTX3 may also exert a significant impact on BC. To this purpose, aggressive/low PTX3-expressing wild-type MB49 cells were grafted in syngeneic transgenic TgN(Tie2-hPTX3) mice, in which PTX3 expression is driven by the endothelial specific promoter Tie2. These animals are characterized by high levels of PTX3 protein in the blood stream and its accumulation in the stroma of different organs [19]. As shown in Figure 6A, endothelial expression and stroma accumulation of PTX3 significantly reduced the growth and Ki67^+^ cell proliferation index of MB49 tumors grafted in transgenic mice when compared to wild-type animals.

To further investigate the impact of stromal PTX3 overexpression under experimental conditions that mimic more closely the natural BC onset and progression, we exploited the carcinogen-induced N-butyl-N-(4-hydroxybutyl)-nitrosamine (BBN) model of BC [20]. Thus, wild type and TgN(Tie2-hPTX3) mice were treated with BBN for 22 weeks by oral gavage. Next, bladders were explanted and analyzed for the presence of hyperplastic areas, carcinoma in situ (CIS), or infiltrating carcinoma at the end of the experimental procedure. Histopathological analysis revealed that PTX3 overexpression delays BBN-induced BC progression towards an aggressive/invasive phenotype. Indeed, the percentage of animals with infiltrating BC was lower in TgN(Tie2-hPTX3) mice in respect to wild-type animals (22% vs. 43%). Conversely, TgN(Tie2-hPTX3) mice showed a higher incidence of carcinoma in situ (44% vs. 29%) (Figure 6B). Accordingly, bodyweight loss at the end of BBN-treatment was significantly higher in wild-type mice in respect to TgN(Tie2-hPTX3) animals (Figure 6B). Interestingly, PTX3 immunostaining in pathological bladders from BBN-treated wild-type mice revealed that PTX3 expression (absent in normal/untreated bladders; data not shown) is present during the in situ carcinoma stage and disappears or is highly reduced when tumors become invasive and more aggressive (Figure 6C). Together, these data further support the hypothesis for an oncosuppressive effect of PTX3 in the preinvasive phases of BC.

Based on the antitumor effect exerted by PTX3, we tested the therapeutic profile of the PTX3-derived FGF trap small molecule NSC12 [19,21] in high grade/low PTX3-expressing BC cells. Like PTX3, NSC12 binds various members of the FGF family and inhibits tumor growth, vascularization, and metastatic capacity in different FGF-dependent tumor models. As shown in Figure 6D, treatment with NSC12 significantly reduced the proliferation rate of 5637 and HT1376 cells (IC_50_ = 5.8 µM and 8.7 µM, respectively) and was accompanied by reduced phosphorylation of FGFR, FRS2, and ERK1/2 proteins in these cells (Figure 6E). 

In vivo, treatment with NSC12 significantly reduced tumor burden of 5637 cells when grafted subcutaneously in NOD/Scid mice (Figure 6F). At variance, NSC12 exerted only a modest effect on the growth of HT1376 grafts, in agreement with the lower in vitro efficacy of the compound on these cells (Figure 6G).

## 3. Discussion

Molecular and genetic studies have been providing novel information about molecular alterations and driver genes for the characterization of subtypes or clusters of BC and the identification of proteins that may serve as biomarkers or therapeutic targets [22,23]. 

Previous observations had shown that the soluble pattern recognition receptor PTX3 is a component of the innate immunity that may exert oncosuppressive functions in different types of tumors [8,10,11,12,19,24,25]. Uroepithelial cells produce PTX3 to prevent urinary infections (such as that due to the uropathogenic *Escherichia coli*), thus contributing to innate resistance to infections in neonates [14,26]. However, its role in BC has never been investigated.

In this study, starting from database analysis and immunehistochemical data on tumor samples from BC patients, we’ve shown that PTX3 downmodulation follows the progression of BC tumors from low-grade to high-grade/muscle invasive BC. These data were confirmed by the analysis of PTX3 expression in tumor cell lines reflecting different grades of BC progression. The modulation of PTX3 expression in these cells by shRNA-mediated silencing or lentiviral-driven overexpression shows that progressive epigenetic silencing of PTX3 facilitates the proliferative potential of BC cells and progressively favors an increased metabolic fitness of cancer cells towards a more glycolytic/aggressive phenotype.

The pleiotropic effects exerted by PTX3 [8] and its dual role as an immune and tissue remodeling player [27] make difficult the identification of the mechanism(s) responsible for the oncosuppressive effect exerted by PTX3 in vitro and in vivo on BC cells. Moreover, the detection of increased levels of “systemic” PTX3, like in the urines of BC patients [15], may be the result of the general inflamed milieu generated by and during tumor progression, thus masking the regulatory effect exerted by PTX3 produced by cancer cells. Our data show that the modulation of PTX3 expression in BC cells is accompanied by a significant impact on the activation of the FGF/FGFR system, which is expressed in BC cells and represents a promising therapeutic target in urothelial cancer [5,28]. Even though we cannot rule out the possibility that other pathways/interactors might be involved and/or affected by PTX3 modulation, experimental data suggest that the FGF trap activity of PTX3 [19], with the consequent inactivation of FGFR signaling, may play a relevant role in this context. Moreover, since BC cells and tumor stroma actively produce FGF ligands, the progressive loss of PTX3 and of its FGF-trap function during BC progression may impact on the cross-talk among different components of tumor microenvironment that drive proliferation, angiogenesis, immune modulation, and drug resistance [29]. 

It is worth mentioning that, at variance with the results obtained with grade II 5637 cells, the effect of PTX3 overexpression in grade III/MIBC HT1376 cells only partially affects their tumor features and aggressiveness, resulting in a very modest inhibitory effect on the growth of tumor xenografts in vivo. These data suggest that mechanisms independent from PTX3 or the PTX3/FGF axis may take over during BC progression, thus explaining the reason why PTX3 restoration is not sufficient to revert the phenotype of these cells and to exert a significant impact on tumor growth. Accordingly, an FGF trap approach represented by treatment with the PTX3-derived small molecule NSC12 was able to inhibit the growth of grade II BC 5637 tumor grafts with only a limited effect on grade III/MIBC HT1376 lesions. From a therapeutic point of view, these data add new information about the use of tyrosine kinase FGFR inhibitors, like BGJ398 (NCT01004224) and Pemigatinib (NCT03914794), which are currently in clinical trials, or Erdafitinib (recently approved by the FDA), for their use on BC patients with alterations of the FGF/FGFR system. 

As already shown for different solid tumors [30], a subpopulation of undifferentiated cells exhibiting stem-like properties has been described also in BC as responsible for enhanced tumor growth, invasiveness, resistance to therapies, and recurrence [31]. While the biological contribution of these cancer stem cells to the growth of tumor is getting clearer, the prognostic significance of these cells in BC is still debated. This is mainly due to the lack of universal markers and of a clear correlation between the presence of cancer stem cells and BC patient outcomes [32,33]. Interestingly, our data indicate that the modulation of PTX3 in BC cell lines impacts on the expression of stemness markers and on stem-like features of these cells. Given the role of FGF2 in the maintenance of cancer stem-like cells [34], the loss of stem-like traits in PTX3 overexpressing BC cells might be the result of the FGF trap action of PTX3, with a possible significant impact on different tumor features. In particular, PTX3 overexpression in high grade/MIBC cells reduces the expression of stemness markers and multidrug resistant ABC-transporters genes are implicated in chemoresistance [35]. Accordingly, preliminary data suggest that PTX3 overexpression/FGF trapping might enhance the sensitivity of these MIBC cells to chemotherapeutics such as methotrexate. Additional studies are required to understand whether a pharmacological anti-FGF/FGFR approach (as provided by FGF traps and/or tyrosine kinase FGFR inhibitors) may also have an impact on drug resistance in BC cells, thus representing a potential/promising combination or adjuvant approach for classical chemotherapy.

## 4. Materials and Methods

### 4.1. Reagents and Cell Cultures

Human RT4, 5637, and HT1376 cells were from American Type Culture Collection (ATCC); murine MB49 cells were a kind gift by Dr. O’Donnell (University of Iowa Hospitals and Clinics, Iowa City, IA, USA). More details in the Appendix A.

### 4.2. Methylation Analysis

Cells (1 × 10^6^) were harvested and gDNA were isolated by ReliaPrep gDNA Tissue Kit (Promega, Milan, Italy). gDNA was sonicated and 250–1000 bp gDNA fragments were used to perform MIRA assay with MethylCollecor Ultra Kit (Active Motif, Carlsbad, CA, USA). Specific primers for PTX3 CpG islands were used (see Appendix A). 

### 4.3. In Vitro Assays

*Cell Proliferation.* Cells were seeded (5 × 10^3^) in 48-well culture plates in complete medium, detached, and counted using the MACSQuant Analyzer (Miltenyi Biotec, Bologna, Italy) after 24, 48, and 72 h. Treatment with NSC12 (from 0.1 to 20 µM) was performed in 1% FBS for 24 h.

*Clonogenic Assay.* Five hundred cells were seeded in 6-well culture plates and incubated in complete growth medium until visible colonies were formed. Then, the supernatant was removed and cells stained with 0.1% crystal violet/20% methanol; plates were photographed (colonies counted using ImageJ software) and solubilized with 1% SDS solution (to measure absorbance at 595 nm). 

*Soft Agar Assay.* Cells (5 × 10^4^) were suspended in 3 mL of complete growth medium containing 0.3% agar and poured on to 2 ml pre-solidified 0.6% agar in a 6-well plate. After 3 weeks of incubation, colonies were observed under a phase contrast microscope, photographed, and their area was measured using the Image J Software and the SA_NJ algorithm [10]. 

*Wound-Healing assay.* Confluent cells were scraped with a 200 µL tip to obtain a 2 mm thick denuded area. After 24 h, wounded monolayers were photographed and the width of the wounds was measured in 3 independent sites per group.

### 4.4. Tumour Sphere Formation Assay and ALDH Analysis

Three hundred 300 cells were resuspended in DMEM/F-12 medium (GIBCO Gaithersburg, MD, USA) containing 10 ng/mL basic fibroblast growth factor (bFGF), 10 ng/mL epidermal growth factor (EGF) and 2% of B27 supplement (Sigma-Aldrich, Milan, Italy) and plated into each well of 24-well Ultra-Low Attachment Plates (Corning, NY, USA). After 7 days of incubation, tumor spheres were counted and assayed for ALDH activity using the Aldefluor kit (Stem Cell technologies, Meda MB, Italy) according to manufacturer’s instructions. ALDH-positive cell analysis was performed by FACS MACSQuant cytofluorimeter.

### 4.5. Seahorse and ATP Quantification

Cells were seeded on Seahorse XFe24 culture plates (Agilent, Santa Clara, CA, USA); oxygen consumption (OCR) and extracellular acidification (ECAR) measurements were performed at 6 min intervals (2 min mixing, 2 min recovery, 2 min measuring) in a Seahorse XFe24 Extracellular Flux Analyzer (XFe Wave software). Consecutive treatments with oligomycin (1 µM final), FCCP (0.5 µM final), and rotenone/antimycin A (0.5 µM final) were performed to enable quantification of basal OCR, ATP-coupled OCR, proton leak, and maximal respiration (Seahorse XF Cell Mito Stress Test, Agilent). ATP production was quantified with ATP Detection Kit (Molecular Probes, Eugene, Oregon, USA) following manufacturer’s instruction.

### 4.6. Immunohistochemistry

For histological analysis for immunohistochemistry, samples were fixed in formalin, embedded in paraffin blocks, and sectioned at a thickness of 3 µM. Processing included dewax in xylene, hydration, and staining with hematoxylin and eosin (H&E). The following primary antibodies were used: Rabbit polyclonal anti-human PTX3, rat monoclonal anti-mouse Ki67 (Dako, Milan, Italy), rabbit anti-phospho-Histone H3 (Ser10) (Millipore, Burlington, MA, USA). For revelation, HRP-labeled polymer anti-rabbit or anti-rat and Vectastain Elite ABC kit (Vector Laboratories, Burlingame, CA, USA) were used. Positive signal was revealed by 3,3′-diaminobenzidine staining (Roche, Monza, Italy) and counterstained with Carazzi’s haematoxylin to identify nuclei, dehydrated, and mounted in DPX (Sigma) before analysis by light microscopy. Acquisition of images was performed with the automatic high-resolution scanner Aperio System (Leica Biosystems, Wetzlar, Germany, EU). 

### 4.7. In Vivo Studies

In vivo were approved by the local animal ethics committee (OPBA, Organismo Preposto al Benessere degli Animali, Università degli Studi di Brescia, Brescia, Italy) and were performed in accordance with national guidelines and regulations. Seven-week-old NOD/Scid and C57BL/6 male mice were injected subcutaneously (s.c.) into the dorsolateral flank with 3 × 10^6^ wild-type, mock, and PTX3-transfected human (RT4, 5637, or HT1376) and 2.5 × 10^5^ murine (MB49) cells. Tumors were measured with calipers and the volume was calculated according to the formula V = (D × d2)/2, where D and d are the major and minor perpendicular tumor diameters, respectively. The volume of tumor grafts was analyzed with a 2-way analysis of variance, and individual group comparisons were evaluated by the Bonferroni correction. At the end of the experimental procedure, tumors were harvested, weighed, and embedded in paraffin for immunohistochemical analysis (see Appendix A). BBN (0.05% in the drinking water) treatment was performed on wild-type and transgenic TgN(Tie2-hPTX3) females from week 8 to week 30 of age [20]. 

## 5. Conclusions

In conclusion, our data demonstrate that the modulation of PTX3 expression in BC cells exerts a significant impact on different biological features of BC growth, including cell proliferation, motility, metabolism, stemness, and drug resistance, thus resulting in an oncosuppressive effect on BC progression. In addition, database analysis and immunehistochemical data on tumor samples from BC patients raise the possibility that PTX3 may represent a potential functional biomarker in BC evolution. Further studies will be required to elucidate this point.

## Figures and Tables

**Figure 1 cancers-11-01277-f001:**
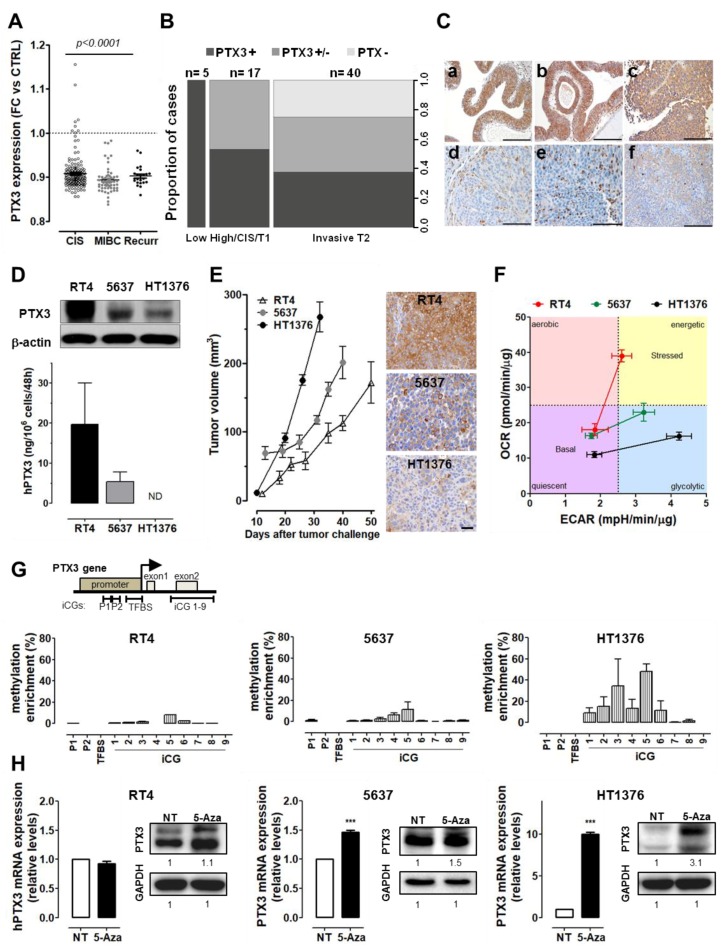
Long pentraxin-3 (PTX3) expression is related to tumor grade in human bladder cancer (BC) samples and tumor cell lines. (**A**) Scatter Plot of PTX3 expression levels in human BC samples compared with normal tissue, data obtained from Gene Expression Omnibus (GEO). (**B**) Spine plot graph showing the association between PTX3 immunostaining and tumor grade in human BC samples *p* = 0.0019; *n* = number of cases. (**C**) Human low-grade papillary bladder carcinomas (LGP-BC; *n* = 3) (**a**–**c**) and high-grade muscle invasive bladder cancer (MIBC; *n* = 3) (**d**–**f**) stained for PTX3. Magnification: 20 ×; scale bar = 100 µM. (**D**) Quantification of PTX3 levels in BC cell lysates (Western Blot, upper panel) or medium (ELISA, lower panel). (**E**) Tumor growth in vivo of RT4, 5637, and HT1376 cells, and IHC for PTX3 of the explanted tumors. scale bar: 50 µM. (**F**) Characterization of the energy metabolism of BC cells by Seahorse Mito Stress Test (OCR = oxygen consumption rate, ECAR = extracellular acidification rate). (**G**) Percentage of methylation enrichment of the PTX3 promoter in BC cell lines (**G**) and quantification (by qPCR and Western blot) of PTX3 expression levels after treatment with 5-Aza-2-deoxycytidine (5-AZA) (**H**). Data are expressed as mean ±SEM. *** *p* < 0.001.

**Figure 2 cancers-11-01277-f002:**
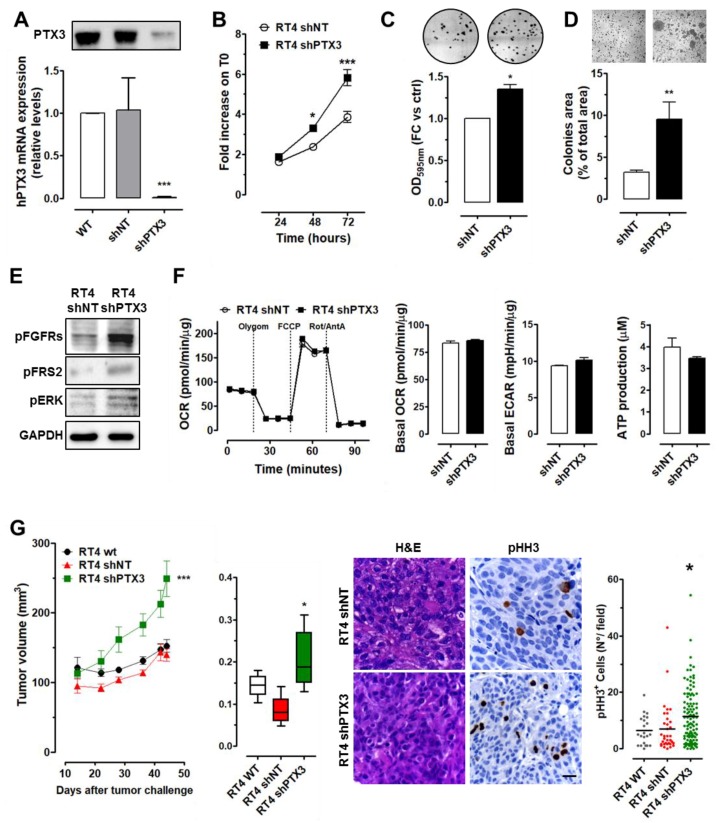
PTX3 silencing in low-grade RT4 BC cells. (**A**) PTX3 expression (mRNA and protein levels) in silenced (shPTX3), control (shNT), and wild type (WT) RT4 cells. Characterization of RT4 cells by cell proliferation (**B**), colony formation (**C**), and soft agar (**D**) assays. (**E**) Western blot analysis of pan-FGFR, FRS2, and ERK phosphorylation. (**F**) Seahorse Mito Stress Test and ATP production analysis performed on RT4-shPTX3 and RT4-shNT cells. (**G**) Tumor growth and weight of RT4 cell xenografts. Hematoxylin and eosin (H&E) and pHH3 immunostaining quantification performed on tumor samples. Magnification: 40×; scale bars: 20 µM. Data are expressed in scatter plot graphs. * *p* < 0.05, ** *p* < 0.01, *** *p* < 0.001.

**Figure 3 cancers-11-01277-f003:**
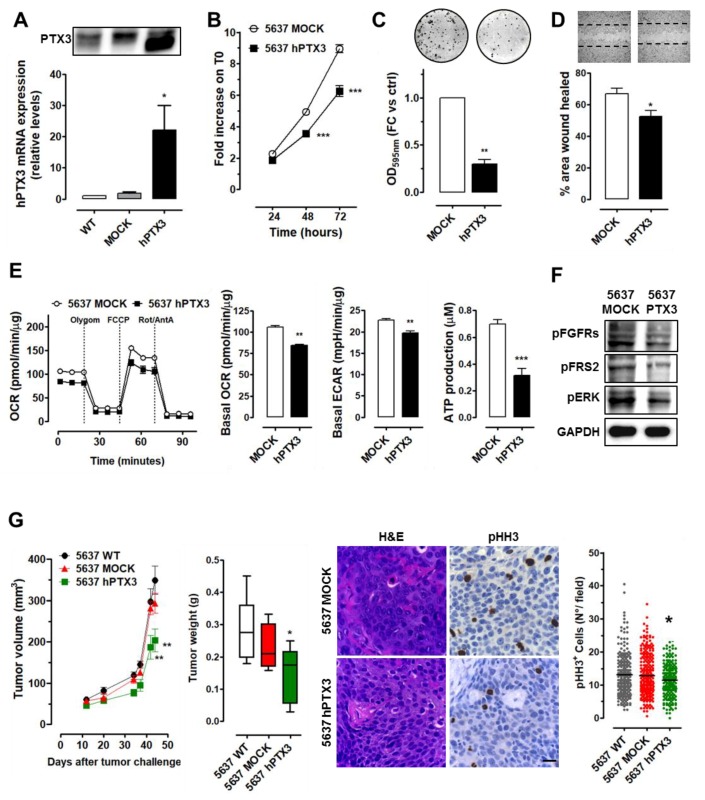
PTX3 modulation in grade II 5637 BC cells. (**A**) PTX3 expression (mRNA and protein levels) in PTX3 transduced (hPTX3), control (Mock), and wild type (WT) 5637 cells. Characterization of 5637 cells by cell proliferation (**B**), colony formation (**C**), and wound repair (**D**) assays. € Seahorse Mito Stress Test (OCR and ECAR) and ATP production analysis performed on 5637-Mock and 5637-hPTX3 cells. (**F**) Western blot analysis of pan-FGFR, FRS2, and ERK phosphorylation. (**G**) Tumor growth and weight of 5637 cell xenografts. H&E and pHH3 immunostaining quantification performed on tumor samples. Magnification: 40×; scale bars: 20 µM. * *p* < 0.05, ** *p* < 0.01, *** *p* < 0.001.

**Figure 4 cancers-11-01277-f004:**
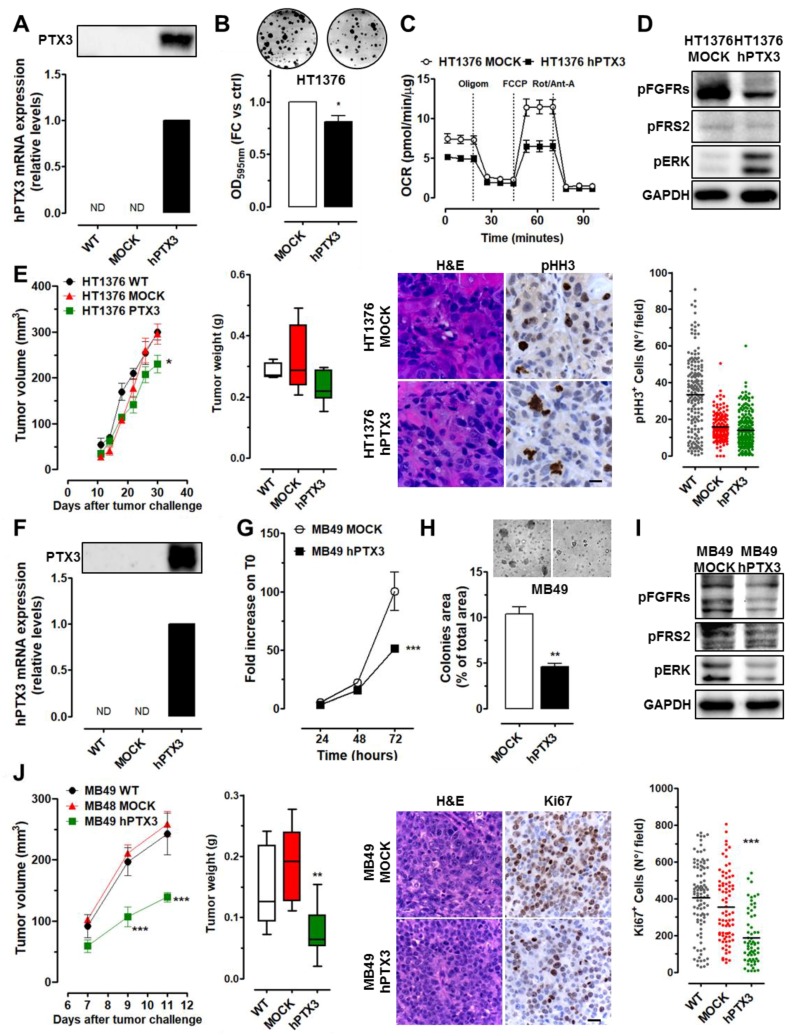
PTX3 modulation in high-grade/invasive HT1376 and MB49 BC cells. (**A**) PTX3 expression (mRNA and protein levels) of PTX3 transduced (hPTX3), control (Mock), and wild-type (WT) HT1376 cells. Characterization of HT1376 cells by colony formation (**B**), Seahorse Mito Stress Test (**C**), and Western blot (**D**) assays. (**E**) Tumor growth and weight of HT1376 cell xenografts, H&E, and pHH3 immunostaining quantification performed on tumor samples. (**F**) PTX3 expression (mRNA and protein levels) of PTX3 transduced (hPTX3), control (Mock), and wild type (WT) MB49 cells. Characterization of MB49 cells by proliferation (**G**), clonogenic/anchorage-independent growth (**H**), and Western blot (**I**) assays. (**J**) Tumor growth and weight of MB49 cell xenografts, (H&E and Ki67 immunostaining quantification performed on tumor samples). Magnification: 40×; scale bars: 20 µM. * *p* < 0.05, ** *p* < 0.01, *** *p* < 0.001.

**Figure 5 cancers-11-01277-f005:**
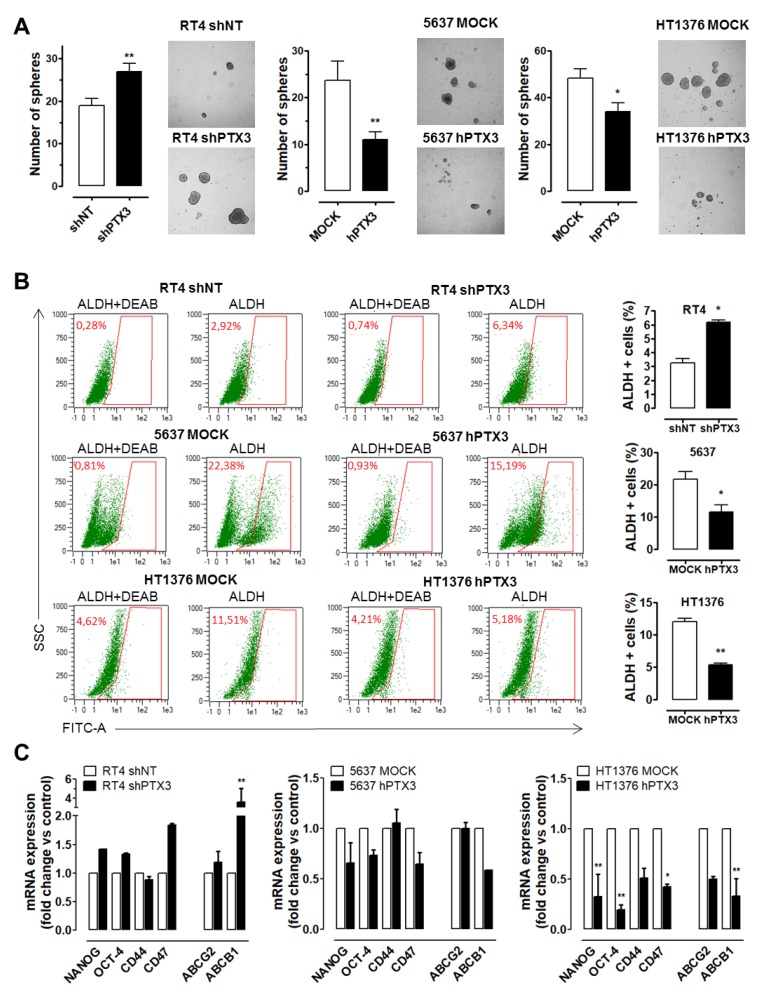
PTX3 modulation affects stem-like features in BC cells. (**A**) Sphere formation assay performed on RT4, 5637, and HT1376 transfectants. Quantification of sphere number (histograms) and representative images are shown. (**B**) Flow cytometry analysis of ALDH activity in modulated BC sphere-forming cells. The ALDH inhibitor DEAB was used as negative control for baseline fluorescence of cells. Histograms show the percentage of ALDH^+^ cells (mean ±SEM, *n* = 3). (**C**) Quantitative RT-PCR analysis of stemness and drug resistance genes in PTX3-modulated BC cells. * *p* < 0.05, ** *p* < 0.01.

**Figure 6 cancers-11-01277-f006:**
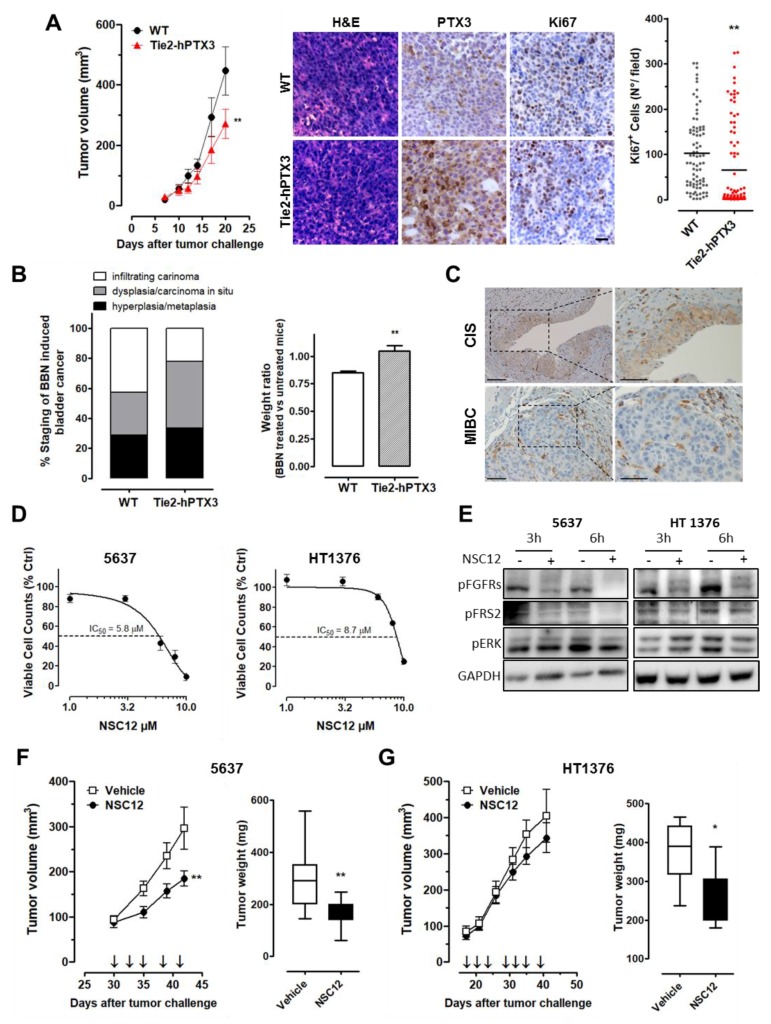
Stromal expression of PTX3 and NSC12 treatment impair BC growth. (**A**) Tumor growth and immunohistochemical analysis of MB49 cells grafted in wild-type (WT) or transgenic TgN(Tie2-hPTX3) mice. scale bar: 20 µM. (**B**) Histopathological evaluation of bladders from WT and TgN(Tie2-hPTX3) mice treated with N-butyl-N-(4-hydroxybutyl)-nitrosamine (BBN). Body weight variation of treated mice is shown (*n* = 10–14 mice/group). (**C**) Representative PTX3 immunostaining of carcinoma in situ (CIS) and muscle invasive BC (MIBC) samples in BBN-treated wild type mice. Magnification: 10× and 20×; scale bar = 100 µM. Viable cell counting (**D**) and Western blot analysis (**E**) of BC cells treated with NSC12. Growth and weight of 5637 (**F**) and HT1376 (**G**) tumor grafts treated with NSC12 or vehicle (*n* = 8 mice/group; * *p* < 0.05, ** *p* < 0.01).

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
