# Peer review of "Long Pentraxin-3 Follows and Modulates Bladder Cancer Progression"

_cancers, 2019, doi:10.3390/cancers11091277_

Round 1

Reviewer 1 Report

In their manuscript „Long Pentraxin-3 follows and modulates bladder cancer progression“, Matarazzo and colleagues demonstrate the impact of PTX3 in bladder cancer.

The manuscript is very well written, structured and despite the complex analyses relatively easy to follow. The work shows interesting data and gives a comprehensive analysis of PTX3 in this setting.

The only point I might have, is that it would be desirable to have analyzed not three but more cell lines of different grades/molecular subtypes/aggressiveness to achieve more solid data (>=two in each group). However, this does not deteriorate the quality of this excellent work.

One minor thing: in line 331, in the discussion section, erdafitinib has already been approved by the FDA earlier this year for use in the US.

Reviewer 2 Report

This manuscript described PTX3 exerts an oncosuppressive effect on bladder cancer (BC) progression by silencing PTX3 in RT4 cells and overexpression PTX3 in 5637 cell. PTX3 modulation affects stemness features in BC cells and stromal expression of PTX3 also hampers BC growth. This is an interesting and elaborate study. However, there are some points need to address or additional remarks to fulfill the manuscript. The terms in the texts and in the figures need consistent. In page 2, line 78 mentioned non muscle invasive lesions (NMIBC), but in the Fig. 1A, author labeled CIS to indicated NMIBC. The full name of carcinoma in situ (CIS) need to show in the article or figure legend. In figure 1A, results did not show the PTX3 expression in recurrence group. Authors need to supply the statistical significance in CIS, MIBC, and recur groups. In figure 1H, it is suggested to label the cell lines in these three figures. Figure 1F and Figure 2F use the same method, but the method need to consistent in name. In figure 1F mentioned Seahorse technology, but in figure 2F is seahorse Mito Stress test. The OCR name (OCR=oxygen consumption rate) need to show in the first appearance in the article in figure 1F not in the second mention in figure 2F. The title of figure 2 and figure 4 are suggested to show the cell line names in as the same in figure 3. In the figure 3A, figure legends or the text do not show the meaning of hPTX3. Is it the same as PTX3 transduced (PTX3)? Reference 15 (A multi-analyte assay for the non-invasive detection of bladder cancer) mentioned the urine analysis of PTX3 concentration in bladder cancer patients (1.1767 ng/ml) and in normal subjects (0.900.89) (p= 0.267). Authors need to discuss the correlation of this finding and the results in present manuscript.

Author Response

Reviewer 2.
The terms in the texts and in the figures need consistent. In page 2, line 78 mentioned non muscle invasive lesions (NMIBC), but in the Fig. 1A, author labeled CIS to indicated NMIBC.
The full name of carcinoma in situ (CIS) need to show in the article or figure legend.

We thank the Reviewer, and we corrected this point in the main text as “non-muscle invasive lesions (NMIBC)/carcinoma in situ (CIS)”.

In figure 1A, results did not show the PTX3 expression in recurrence group. Authors need to supply the statistical significance in CIS, MIBC, and recur groups.

If the Reviewer refers to Figure 1A, the statistics for each group (CIS, MIBC and recurrence) have been grouped and indicated as <0.0001 for the ease of comprehension of the Figure. Single statistical values <0.0001 were not specified by the software
If the Reviewer refers to Figure 1B, while data in Figure 1A were obtained from online database (GEO), these were data from samples collected by the Unit of Pathology and only included these three histotypes (low grade, NMIBC/CIS and invasive) that for instance are the ones represented by cell lines in the rest of the study. The statistical significance for the association between PTX3 immunostaining and tumor grade (p=0.0019) has been reported in the legend to avoid “overload” of the Figure panel. In case the Reviewer deems it appropriate we can add this value in the graph.

In figure 1H, it is suggested to label the cell lines in these three figures.This has been corrected in the Revised Figure 1

Figure 1F and Figure 2F use the same method, but the method need to consistent in name. In figure 1F mentioned Seahorse technology, but in figure 2F is seahorse Mito Stress test. The OCR name (OCR=oxygen consumption rate) need to show in the first appearance in the article in figure 1F not in the second mention in figure 2F.

This has been corrected in the revised version of the manuscript.

The title of figure 2 and figure 4 are suggested to show the cell line names in as the same in figure 3. In the figure 3A, figure legends or the text do not show the meaning of hPTX3. Is it the same as PTX3 transduced (PTX3)?
Regarding the title of Figures 2 and 4, this has been corrected. Regarding hPTX3 this is the transduced PTX3 (human gene), the discrepancy has now been corrected along the paper reporting always hPTX3 in the text and in the legends.

Reference 15 (A multi-analyte assay for the non-invasive detection of bladder cancer) mentioned the urine analysis of PTX3 concentration in bladder cancer patients (1.1767 ng/ml) and in normal subjects (0.900.89) (p= 0.267). Authors need to discuss the correlation of this finding and the results in present manuscript.

The Reviewer is right, we have now explained this aspect in the Discussion (line 318) reporting that the systemic/urine presence of PTX3 is often the result of a general inflamed context that is generated during and by tumor growth. In contrast the local PTX3 produced by tumor cells (as explained by our data) is responsible for the oncosuppressive effect observed in vitro and in vivo.

I would like to thank the Reviewers for their comments that have improved the quality of our work.
Hoping to have fulfilled the requirements for publication,
I remain yours sincerely,